# OpenReview forum: "Evaluating Large Language Models' Capability to Conduct Cyberattacks On Embedded Devices"
_ICLR.cc/2025/Conference — Submitted to ICLR 2025_

### Official Review · Reviewer_eVAV · 2024-11-02

**Soundness:** 2
**Presentation:** 1
**Contribution:** 2
**Rating:** 3
**Confidence:** 4

**Summary:**

This work focuses on using LLMs to generate exploit targeting against vulnerabilities on embedding devices. The author conducted 32 cyberattacks on five devices and evaluated the performance based on DREAD framework.

**Strengths:**

1. The authors conducted cybersecurity attacks on real-world devices.
2. Detailed documentation is provided in the supplementary materials.

**Weaknesses:**

While we recognize that the page limit may have restricted the authors from providing extensive experimental details, the paper lacks essential information on the cyberattacks involving LLMs.

1. The paper does not explain the reason behind selecting these 32 specific attacks as representative experiments for LLMs.
2. The authors should elaborate on the scoring metrics used within the DREAD framework.
3. In addition to presenting DREAD framework scores, the authors should clarify the attack performance on target devices.
4. The authors mention on Page 13 that the code is available in a GitHub repository, but there is currently no code uploaded to the specified repository.

**Questions:**

Please refer to "Weaknesses."

---

### Official Review · Reviewer_YwT8 · 2024-11-04

**Soundness:** 2
**Presentation:** 2
**Contribution:** 2
**Rating:** 3
**Confidence:** 4

**Summary:**

The paper explores the capabilities of large language models (LLMs) in conducting cyberattacks on consumer-grade embedded devices. The authors conduct 32 distinct attacks on five connected devices, including door locks and smart vehicle adapters, comparing manual execution with LLM-assisted methods. Utilizing the DREAD framework, they assess the effectiveness of LLMs in increasing reproducibility, exploitability, and discoverability, noting that while LLMs assist in making attacks more accessible, they do not escalate damage or affected user count. Such results highlight potential benefits of LLMs for red-teaming.

**Strengths:**

- The research idea of evaluating the capabilities of LLMs in attacking embedded devices is interesting and important.
- Multiple metrics are considered when evaluating the attacks, such as damage, reproducibility, exploitability.

**Weaknesses:**

**Overview:**

- Lack of clear definition for the threat model.
- The focus on five devices, while practical, may limit generalizability. Additional device categories would strengthen the argument.
- The LLMs still require significant manual input in numerous cases, which weakens the claim of autonomy. The study could better specify the nature and frequency of interventions.
- The study could benefit from a comparative analysis of LLM capabilities versus traditional automation and scripting tools.
- Lack of ethical discussion in the paper.
- The paper violates the double-blind policy by disclosing the institution and author information in the shared GitHub repository.


**Details**:

- The paper lacks a structured threat model, making it unclear what level of attacker capability or access is assumed. Specifying the assumed attacker goals, resources, and access level would provide essential context for interpreting the effectiveness and limitations of LLM-assisted attacks.

- The study’s focus on five consumer IoT devices limits its generalizability. Expanding to a wider array of devices—such as wearables or medical IoT—using various protocols (e.g., Zigbee, Bluetooth Low Energy) could reveal additional insights about LLM effectiveness across diverse architectures and security challenges.

- The autonomy scores indicate varied dependence on manual input across attacks. Breaking down when and why LLMs require intervention, for example, syntax correction versus domain-specific attack knowledge, would clarify the scope of LLM effectiveness in different cybersecurity tasks (e.g., reconnaissance, code generation).

- The study notes that LLMs failed when domain-specific information was missing, such as device firmware specifics. A clearer categorization of failure types (e.g., knowledge gaps, complex scripting) would offer practical insights into LLM limitations, guiding future improvements, such as incorporating more up-to-date technical knowledge in training data.

- A comparison with traditional automation security tools (e.g., Metasploit, Nmap) would add context, showing where LLMs uniquely contribute or fall short. Highlighting scenarios where LLMs outperform or underperform these tools would help clarify their value in cybersecurity applications.

- The authors use prompt engineering to bypass LLM guardrails, showing how easily LLMs might be guided to generate harmful content under ethical pretexts. Expanding on the ethical and security implications, along with suggestions for improved guardrails, would add value to the study’s findings on LLM safety mechanisms.

- I appreciate the authors' commitment to open science by providing a GitHub repository with supplementary materials. However, some documents within the repository disclose the institution and author information, potentially violating the double-blind review policy.

**Questions:**

- Can you clarify the assumed threat model, including the attacker's capabilities, resources, and access levels?
- What criteria did you use to select the devices and attack types, and how do you see this selection impacting generalizability?

**Details Of Ethics Concerns:**

- The authors use prompt engineering to bypass LLM guardrails, showing how easily LLMs might be guided to generate harmful content under ethical pretexts. Expanding on the ethical and security implications, along with suggestions for improved guardrails, would add value to the study’s findings on LLM safety mechanisms.

- I appreciate the authors' commitment to open science by providing a GitHub repository with supplementary materials. However, some documents within the repository disclose the institution and author information, potentially violating the double-blind review policy.

---

### Official Review · Reviewer_9fEE · 2024-11-04

**Soundness:** 1
**Presentation:** 3
**Contribution:** 1
**Rating:** 3
**Confidence:** 4

**Summary:**

This paper answers the question "Can LLMs be used to hack devices?" Specifically, with the help of 32 simulated attacks on 5 embedded devices and 1 cyber attack game, this paper compares  blue team (ethical hackers / penetration testers) successfully carrying out an attack (a) manually (b) with LLM assistance. Each attack (a) and (b)  is scored using one of the standards 5-metric DREAD score (on a scale of 0 to 10).

The findings reveal how easy it is to attack a device and what assistance was provided by the ChatGPT (v3.5 and v4) LLM. The main contributions of this paper are:
1) Provide a benchmark to evaluate LLM's effectiveness in hacking devices
2) Investigate the ease of hacking a device using LLM (with the implication that a non cyber security expert can use this tool to successfully carry out attacks) which can in turn be used to evaluate the resilience of devices and their susceptibility to cyber attacks.
3) Examples are provided for how to modify the prompts to overcome the security guardrails of LLMs

This paper provides some empirical evaluations on the ongoing research question, "Have LLMs lowered the entry barrier for cyber attackers"

**Strengths:**

1. Using LLMs for Cyber attacks and hacking devices, helping Pentesters is a useful study in cybersecurity domain
2. Some of the findings of this paper (ease of using LLM), contribution factor of LLMs can be used to benchmark
3. Appendix sections provide good background reading (summarized hacking reports)

**Weaknesses:**

The paper is well written but is missing some essential content which has lowered its score. It could be strengthened in the following ways:
1. Evaluating with multiple LLMs. Since the objective of the paper is to evaluate LLMs' capability to conduct cyberattacks on embedded devices, it would benefit from evaluating the effectiveness of multiple LLMs than only one used in the paper - ChatGPT.
2. It is not very clear how the evaluation (DREAD scoring) is done (Section 4). That is number of annotators / evaluators. Did multiple people evaluate the same attacks? If yes, could you talk about inter-annotator agreement?
3.. Overall, while the paper talks about an important topic, its contribution is lessened since the experiments and evaluation could have been more in depth with more LLMs. Currently, the claims need more evidence (more LLMs, more details about te difference between manual and with LLM attacks) to be supported.

**Questions:**

1. The link cited several times in footnotes (https://anonymous.4open.science/...) claims to contain prompt examples, code, and data. Perhaps I am missing something but all I could find was some pdf files of technical reports and publications.
2. In the evaluation, How many evaluators were used? There is some confusion as to whether the testers who attacked were different from the evaluators. Could you please clarify?
3. Perhaps it would be useful for you to cite this paper: https://arxiv.org/html/2402.06664v1 using LLMs to hack websites. The Related work section contains a distinct lack of approaches investigating LLMs used to assist in hacking.

**Details Of Ethics Concerns:**

The paper is giving a recipe on how to use LLMs to attack systems. While it is true that it is for penetration testing (ethical hacking) it could easily be used by an attacker.  Please not this is a MINOR concern because such information is readily available in blogs and openly on the web - how to circumvent the security guardrails on a LLM.

---

### Meta-Review · Area_Chair_JhnR · 2024-12-21

**Metareview:**

The submission titled "Evaluating Large Language Models' Capability to Conduct Cyberattacks On Embedded Devices" provides a series of case reports in cybersecurity where attacks were performed either with or without the assistance of a LLM. Overall, this is an interesting applied study, but the relevance of this submission to ICLR is not clear to me. These reports are certainly interesting, but would be a better fit for another conference. As described by the reviewers, there is also the impression that the study could be improved by systematizing the study, and going beyond case reports to more general claims about LLM-assisted attacks (e.g. in relationship to model scale, or model capability). Overall I do not recommend acceptance for this submission.

I do think there was good, cybersecurity-relevant feedback provided on this draft, especially by reviewer YwT8, which I hope the authors take into account when revising their material.

There are also minor concerns about the disregard of formatting, impossibility to read the paper without refering to the appendix, and minor deanonymization issues, which I have not included in my rating.

**Additional Comments On Reviewer Discussion:**

The authors answer a number of answerable reviewer questions in their author responses, but are unable to address key concerns.

---

### Decision · Program_Chairs · 2025-01-22

Reject